# Revealing Long-context Potential of Attention Heads via Frequency Kernels

**Senyu Han** [1]   **Yilu Cao** [1]   **Kai Yu** [1 3 4]   **Lu Chen** [1 2 3 4]

## Abstract

Large language model (LLM) exists a subset of attention heads that are highly responsible for long-context processing. Existing work has identified different long-context heads in models, but their detection methods mainly rely on model inference on actual long texts and do not analyze the inherent properties of the head parameters. In this paper, we use kernel methods to analyze static *frequency kernels* formed by different rotation frequency components of attention heads, and we design a Long-context Potential Score (LPS) to measure the potential of attention heads in processing long contexts. Kernels of heads with high LPS exhibit concentrated low-frequency energy and low effective rank, which allow them to effectively capture highly specialized information from distant contexts. Experiments and analysis on long-context tasks and model behaviors show that the LPS metrics can well reflect the actual capability of heads on long contexts. Furthermore, by simply amplifying low-frequency kernels of heads with high retrieval potential, we can further improve model's performance on long-context tasks. Our metrics and head enhancement methods are fully static and offline, and they can be quickly conducted under low-resource constraints. Code is publicly available at here.

## 1. Introduction

Large language models (LLMs) based on the Transformer architecture (Vaswani et al., 2017) have achieved strong performance on a wide range of natural language processing tasks. A core component of these models is the self-attention mechanism: it allows the model to dynamically select and combine information from the input context, and multi-head attention (MHA) further enables the model to attend to different aspects of the context at the same time. While the attention mechanism can attend to arbitrarily distant context in theory, the model often suffers from performance degradation as the context length increases (Chiang & Cholak, 2022; Du et al., 2025). This issue can be caused by several factors, including attention dilution (Hsieh et al., 2024b), limited effective context utilization (An et al., 2025), and the difficulty of propagating information across long distances (Laban et al., 2025). As a result, models may fail to retrieve or emphasize important information that appears far from the current token. How to effectively handling long contexts still remains a major challenge for large language models.

While many studies try to propose new architectures to improve model's performance on long contexts, another line of work focuses on analyzing and interpreting the model's existing long-context processing ability. As the attention mechanism is responsible for retrieving key information from long contexts and assigning different importance weights, these works mainly focus on the self-attention layers of LLMs. One notable work made by Wu et al. (2025) identified "retrieval heads", which is a small subset of attention heads that play a critical role in capturing long-range dependencies. This finding has been validated by many subsequent studies and further developed into various strategies to enhance the model's long-context performance (Zhang et al., 2025; Peng et al., 2025; Zhao et al., 2025). Some work also investigates other long-context related heads, such as "induction heads" (Olsson et al., 2022) and "mover heads" (Yao et al., 2024). Such works typically identify long-context heads by analyzing the model's attention distribution or ablation behavior on hand-made inputs or real long-context tasks like Needle-in-a-Haystack (NIAH). Since these methods require running the model on long inputs, it can be very expensive and resource-intensive to identify these heads. Moreover, these methods are usually task-dependent, and they do not directly probe into the intrinsic, inherent parameter properties of the attention heads.

To this end, we aim to identify potential long-context attention heads only by their special properties inside the static parameter. More specifically, the static parameters of an attention head can be seen as a kernel matrix, which is obtained by summing up **frequency kernels** formed by query

[1]X-LANCE Lab, School of Computer Science, Shanghai Jiao Tong University, Shanghai, China [2]Shanghai Innovation Institution, Shanghai, China [3]Jiangsu Key Lab of Language Computing, Suzhou, China [4]Suzhou Laboratory, Suzhou, China. Correspondence to: Lu Chen <chenlusz@sjtu.edu.cn>.

*Proceedings of the $43^{rd}$ International Conference on Machine Learning*, Seoul, South Korea. PMLR 306, 2026. Copyright 2026 by the author(s).

and key components at different positional embedding frequency dimensions (Figure 1). Through theoretical analysis and experimental validation, we confirm that long-context heads exhibit the following properties in their kernels: (1) **The energy of frequency kernels is clearly concentrated in the low-frequency band.** Under the settings of RoPE rotation frequencies, a lower rotation frequency in positional embedding indicates that the corresponding parameters are less sensitive to relative positions, and thus are more likely to consistently capture key signals from distant context. (2) **The attention head kernel have a clearly lower effective rank.** A lower effective rank means that the head kernel is dominated by few frequency components and has a dimensionality bottleneck, and it maps the input queries and keys into a narrower subspace. As a result, these heads are more selective to input features and respond strongly only to specific context information.

Using only static parameters, by jointly considering these two properties, we design a Long-context Potential Score (LPS) to measure the potential of an attention head to process long contexts. Experiments on multiple long-context tasks show that the potential score of an attention head is closely related to its actual long-context capability. Analysis of the actual behavior of high-LPS heads in the model identifies two different functional types of attention heads, namely retrieval and induction. We also analyze the contribution of frequency kernels to the final attention and confirm the existence of attention sinks (Xiao et al., 2024) caused by extremely low-frequency kernels. Furthermore, inspired by these two properties of long-context heads, we propose a lightweight enhancement trick by amplifying the magnitude of low-frequency kernels of the identified retrieval heads. This simple but effective strategy directly improves model performance on long-context tasks. Our scoring and enhancement method is fully static, requires no additional training or inference, and can be completed within seconds.

Our contributions can be summarized as follows:

- We use kernel method to analyze attention heads from static model parameters, and we propose a static scoring metrics to measure the long-context potential of heads without actual inference (Section 3).

- We conduct qualitative attention analysis and quantitative ablation experiments on heads with high LPS. Results show that these heads are indeed crucial for the model's long-context capability, and we further identify their behaviors in the attention (Section 4).

- Based on the properties of long-context heads, we amplify the frequency kernels of heads with high retrieval potential. Experimental results show that this simple enhancement strategy improves long-context processing performance across multiple tasks (Section 5).

## 2. Related Work

**Long-context functionality in attention heads** Previous work has found that different attention heads encode information related to specific functions (Zheng et al., 2024; Todd et al., 2024). These attention heads form functional pathways inside the model that enable the model to perform multiple functions (Yao et al., 2024; Han et al., 2025). For long-context heads, Wu et al. (2025) identify "retrieval heads" by probing their attention distribution on the NIAH task, Zhang et al. (2025) further propose query-focused heads to generalize retrieval heads into broader long-context tasks, and Lee et al. (2025) design complex strategies to boost long-context performance of these heads. Other work identifies "in-context learning heads" (Todd et al., 2024; Yin & Steinhardt, 2025), "induction heads" (Olsson et al., 2022) and more long-context related heads. Almost all these work investigates attention heads by designing specific inputs or tasks and probing their behavior during inference, ignoring the inherent properties inside the head parameters. In contrast, our work starts from the properties that long-context heads should have, and we can identify potential long-context heads without performing actual inference.

**Kernel methods on attention heads** A kernel is a function that measures relationships between data points by implicitly mapping them into a higher-dimensional feature space. Prior research has explored kernel perspectives on the attention mechanism by examining how similarity computations between queries and keys can be interpreted and manipulated (Tsai et al., 2019; Teo & Nguyen, 2024). Beyond reinterpretation, kernelization can also be used to improve efficiency and stability (Luo et al., 2021). Chiang & Yogatama (2025) use RoPE kernels to support the claim that high frequency components may cause dimension inefficiency on long-context tasks, but they do not investigate the properties of the whole head. Besides, their method identifies heads with low dimension utility by training sparse masks, whereas we rely on a purely static parameter analysis approach to detect long-context heads. Recently, Xiong et al. (2025) proposed DoPE, an attention kernel analysis method to suppress noise in the low-frequency components of RoPE-based models and improve in-context learning performance. While our method is similar, we oppositely utilize and enhance these low-frequency head kernels to improve the model's performance on much longer contexts.

## 3. Methodology

In this section, we first introduce the computation of our head kernel and frequency kernel method, then present two scores to measure kernel properties that are necessary for long-context modeling. Finally, we derive our long-context potential score using these properties.

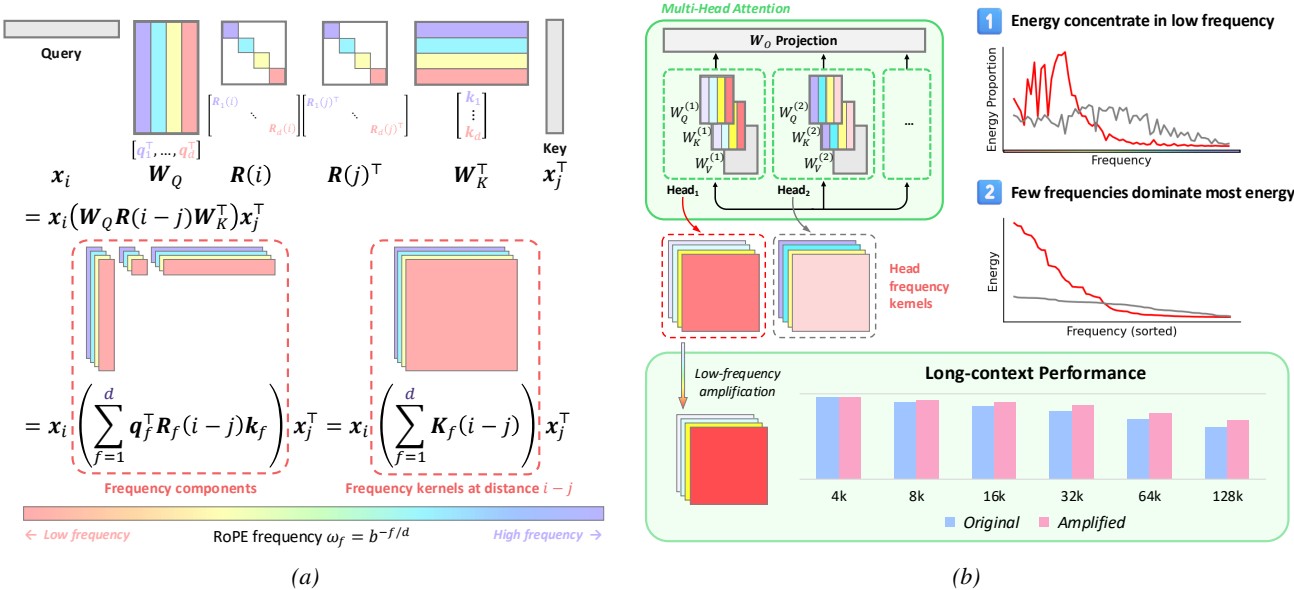

*Figure 1.* (a) By pre-computing the kernels of attention heads at different RoPE frequencies and distances, we can analyze the properties of heads without actual inference on long contexts. (b) Kernels of some heads have special properties that are more suitable for processing long contexts. Amplifying their low-frequency components can further improve the model's performance on long-context tasks.

### 3.1. Preliminary: RoPE and Frequency Kernels

Our work mainly focuses on decoder-only transformers with RoPE (Su et al., 2024) and without bias terms in query or key. For a certain multi-head attention layer, we denote the input hiddens at absolute position $i$ and $j$ as row vectors $\boldsymbol{x}_i \, \boldsymbol{x}_j \in \mathbb{R}^h$. With RoPE, the query and key states of a certain head can be computed as

$$\boldsymbol{Q}_i = \boldsymbol{x}_i \boldsymbol{W}_Q \boldsymbol{R}(i), \quad \boldsymbol{K}_k = \boldsymbol{x}_j \boldsymbol{W}_K \boldsymbol{R}(j), \qquad (1)$$

where $\boldsymbol{W}_Q, \boldsymbol{W}_K$ is the head weight with size of $d_h$, and $\boldsymbol{R}(n)$ is the rotation matrix at position $n$. For the head size $d_h$, we have

$$\boldsymbol{R}(n) = \mathrm{diag}\left(\boldsymbol{R}_1(n), \cdots, \boldsymbol{R}_d(n)\right), \qquad (2)$$

where $d = d_h/2$ is the number of frequency pairs, and

$$\boldsymbol{R}_f(n) = \begin{bmatrix} \cos\theta_f(n) & -\sin\theta_f(n) \\ \sin\theta_f(n) & \cos\theta_f(n) \end{bmatrix} \qquad (3)$$

is the $2 \times 2$ rotation matrix with the rotation angle $\theta_f(n) = \omega_f n, \omega_f = base^{-f/d}$. Following the $d = d_h/2$ notion, the attention score from $i$ to $j$ before normalization $(/\sqrt{d_h})$ and softmax can be computed as (Figure 1):

$$\begin{aligned} \boldsymbol{Q}_i \boldsymbol{K}_j^\top &= \boldsymbol{x}_i \boldsymbol{W}_Q \boldsymbol{R}(i) \boldsymbol{R}(j)^\top \boldsymbol{W}_K^\top \boldsymbol{x}_j^\top \\ &= \boldsymbol{x}_i \left(\boldsymbol{W}_Q \boldsymbol{R}(i-j)^\top \boldsymbol{W}_K^\top\right) \boldsymbol{x}_j^\top \\ &= \boldsymbol{x}_i \left(\sum_{f=1}^{d} \boldsymbol{q}_f^\top \boldsymbol{R}_f(i-j) \boldsymbol{k}_f\right) \boldsymbol{x}_j^\top, \end{aligned} \qquad (4)$$

where $\boldsymbol{q}_f, \boldsymbol{k}_f$ is the $2 \times h$ frequency block of $\boldsymbol{W}_Q, \boldsymbol{W}_K$, indexed as $\left[\boldsymbol{W}_{Q:,2f}, \boldsymbol{W}_{Q:,2f+1}\right]^\top, \left[\boldsymbol{W}_{K:,2f}, \boldsymbol{W}_{K:,2f+1}\right]^\top$. Therefore, the kernel of attention score at relative distance $n = i - j$ and frequency $f$ can be written as

$$\boldsymbol{K}_f(n) = \boldsymbol{q}_f^\top \boldsymbol{R}_f(n) \boldsymbol{k}_f. \qquad (5)$$

For a large positive $base$, we can infer that a larger index $f$ corresponds to a lower frequency $\omega_f$, and the rotation matrix will change less with respect to $n$. By setting different rotation frequencies, positional encoding allows different dimensions within a head to have different distance sensitivities. Dimensions with low frequencies inherently have better ability to retrieve information from the global context.

### 3.2. Effective Rank of Kernels

Effective rank measures how evenly the energy is spread across dimensions in a matrix (Roy & Vetterli, 2007). Originally defined based on the singular values of a matrix, effective rank is computed from the Shannon entropy of their normalized distribution. In our frequency kernel settings, an attention head kernel $\boldsymbol{K}(n)$ is the sum of its frequency kernels $\boldsymbol{K}_f(n)$, and each frequency kernel is responsible for extracting different information from query and key hiddens. If only a small number of frequency kernels contain very large energy, the attention score is then dominated by these few frequencies, leading to a lower effective rank of the head kernel. From this perspective, we compute the effective rank of a head kernel $\boldsymbol{K}(n)$ using the energy of

frequency kernels as:

$$\text{eRank}\left(\boldsymbol{K}(n)\right) = \exp\left(-\sum_{f=1}^{d} p_f \log p_f\right), \quad (6)$$

where

$$p_f = \frac{\|\boldsymbol{K}_f(n)\|_F^2}{\sum_{i=1}^{d} \|\boldsymbol{K}_i(n)\|_F^2}. \quad (7)$$

From a feature extraction viewpoint, head kernels with low effective rank focus more on extracting and amplifying specific features. This property is crucial for retrieving key information from very long, noisy contexts.

### 3.3. Measuring the Long-context Potential of Heads

Attention heads with higher long-context potential should allocate more energy to distance-insensitive low-frequency components, and use these low-frequency kernels to extract key information from the long context. Starting from these two properties, we propose two scoring methods for attention head's long-context potential:

**Frequency energy distribution score** We reward attention heads whose frequency kernel energy is concentrated in the low-frequency band. For a frequency index $f$ (where a larger $f$ indicates a lower frequency $\omega_f$) and its energy proportion $p_f$ according to (7), we compute the weighted energy distribution score as a weighted average:

$$s_{\text{dist}} = \sum_{f=1}^{d} \frac{f p_f}{d}. \quad (8)$$

**Effective rank score** According to (6), we reward attention heads whose kernels have low effective rank:

$$s_{\text{eRank}} = 1 - \frac{\text{eRank}(\boldsymbol{K}(n))}{d}. \quad (9)$$

By jointly considering these two scores, we compute a Long-context Potential Score (LPS) for each attention head:

$$s_{\text{LPS}} = s_{\text{dist}}^{\alpha} s_{\text{eRank}}^{(1-\alpha)}, \quad (10)$$

where $\alpha$ is a weighting coefficient ($\alpha = 0.4$ in our practices). Using LPS on Llama-3.2-3B-Instruct, we identified some top heads that exhibit our desired properties (Figure 2).

By computing and ranking LPS of all attention heads in the model at a long distance $n$, we can quickly identify the heads with higher potential for long-context processing, without performing actual inference on long inputs. In practice, we found that the energy properties of the kernel hardly change with the rotation angle $\theta_f(n)$. Therefore, we fix $n = -32768$ for the following analysis and experiments.

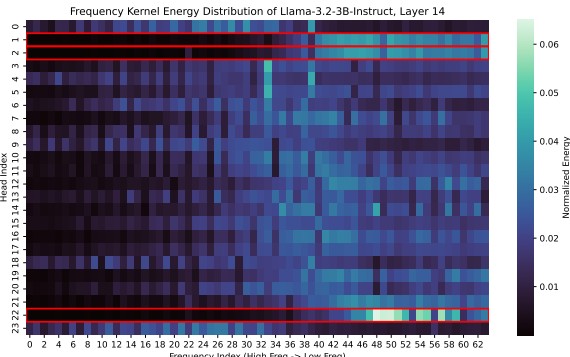

*(a)* Energy distribution heatmap of frequency kernels. The frequency kernel energy of each head is normalized.

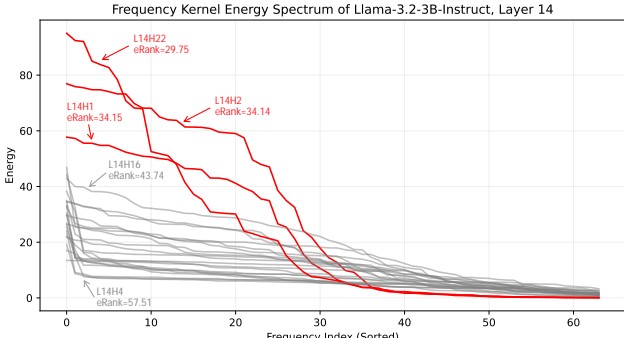

*(b)* Energy descending curves of frequency kernels in each head. We annotated the effective ranks (eRank) of some heads.

*Figure 2.* Visualization of frequency kernel energy distribution in the 14th layer of Llama-3.2-3B-Instruct. The long-context heads we identified (highlighted in red) show distinct properties from other heads: (2a) higher energy concentration on low frequencies and (2b) lower effective rank.

## 4. Frequency Kernels Entail Long-context Attention Heads

In this section, we validate and analyze the attention heads with long-context potential that are identified by frequency kernels. We first verify that these attention heads found by static methods are indeed critical to the model's long-context capability. Then, we analyze the behavior of these attention heads in long contexts to further confirm their practical role. We mainly use Llama-3.2-3B-Instruct, Llama-3.1-8B-Instruct (Grattafiori et al., 2024) and no-thinking Qwen3-4B (Yang et al., 2025) to conduct our experiments.

### 4.1. Verifying Long-context Potentials

Through the following two experiments, we verify that the properties measured by LPS are highly overlapping with the properties of actual long-context heads.

**High-potential heads are important in long-context scenarios.** A direct way to verify that certain attention heads have specific functions is to remove them from the model

*Table 1.* Head masking experiments on RULER (context length from 4K to 128K) and general datasets (MMLU: Hendrycks et al. (2021), HellaSwag: Zellers et al. (2019), ARC-Challenge: Clark et al. (2018), OpenBookQA: Banerjee et al. (2019), GSM8K: Cobbe et al. (2021)). Masking heads with top LPS score will lead to evident performance drop in long-context tasks. Wavy line indicates the worst result. All results are evaluated by OpenCompass (Contributors, 2023).

| Model | RULER | | | | | | General | | | | |
|---|---|---|---|---|---|---|---|---|---|---|---|
| | 4K | 8K | 16K | 32K | 64K | 128K | MMLU | HellaSwag | ARC-C | OBQA | GSM8K |
| **Llama-3.2-3B-Instruct** | 92.79 | 84.72 | **83.67** | **79.47** | 71.07 | 65.78 | **62.14** | **63.15** | 74.24 | **77.00** | **76.95** |
| - w/o 10 random heads | **93.64** | **85.06** | 83.49 | 79.05 | **72.49** | **66.49** | 61.76 | 62.07 | 72.20 | **77.00** | 76.35 |
| - w/o 10 top LPS heads | 66.81 | 52.28 | 39.46 | 30.17 | 17.19 | 11.13 | 61.05 | 63.05 | **74.92** | 76.60 | 69.83 |
| **Llama-3.1-8B-Instruct** | 96.18 | 94.23 | 92.28 | 85.50 | 84.48 | 75.81 | 69.06 | 76.83 | 83.05 | **80.60** | **84.15** |
| - w/o 20 random heads | 95.56 | 93.26 | 92.10 | 84.98 | 83.10 | 74.70 | 65.19 | 68.37 | **82.37** | 80.40 | 78.77 |
| - w/o 20 top LPS heads | 50.18 | 37.98 | 33.34 | 23.74 | 14.40 | 7.93 | 67.17 | 68.86 | 81.69 | 78.60 | 76.95 |
| **Qwen3-4B** | **95.08** | **91.47** | **89.64** | 86.85 | 75.66 | 66.3 | 66.69 | 77.83 | 82.71 | 78.00 | **42.15** |
| - w/o 20 random heads | 94.09 | 88.88 | 84.92 | 83.36 | 70.37 | 61.54 | 64.98 | 74.92 | 77.63 | 75.00 | 41.02 |
| - w/o 20 top LPS heads | 28.5 | 22.81 | 18.02 | 14.79 | 11.28 | 7.61 | 64.97 | 77.13 | 79.32 | **78.20** | 40.18 |

and observe the model's performance on related tasks after losing these heads. In our setting, we use frequency kernel analysis and LPS to identify heads with top scores. After setting their contributions to the residual stream to zero, we then evaluate the performance of the modified model on long-context tasks. Experiments on NeedleBench V2 (Li et al., 2025) and RULER (Hsieh et al., 2024a) show that: After removing these top heads, the model's performance on these tasks experiences severe degradation (Figure 3, Table 1). Additionally, to verify that these heads mainly work in long-context scenarios, we evaluated the masked model on several general commonsense tasks. Results in Table 1 show that masking these long-context heads does not cause really significant performance loss on these tasks, except for the GSM8k dataset on Llama models. Solving the math problems in GSM8k requires the model to perform chain-of-thought reasoning, which demands stronger long-context processing ability. After removing these potential long-context heads, the model's performance on GSM8k drops noticeably, which is consistent with the findings of Wu et al. (2025).

**Long-context (retrieval) heads have high potential scores.** While potential long-context heads indeed have a significant impact on actual long-context scenarios, we also want to examine whether the attention heads identified by our static frequency kernel method are consistent with the long-context heads identified by actual long-context inference. Wu et al. (2025) has proposed *retrieval score* to identify retrieval heads from model inference. We treat heads identified by these online methods as gold labels, and we investigate the overlap between them and our top heads that are statically identified by *retrieval potential score* (a metric derived from LPS, introduce later in Equation 12). Table 2 shows that about or nearly half of the retrieval heads identified by online inference overlap with heads identified by our static method, and this overlap is much higher than

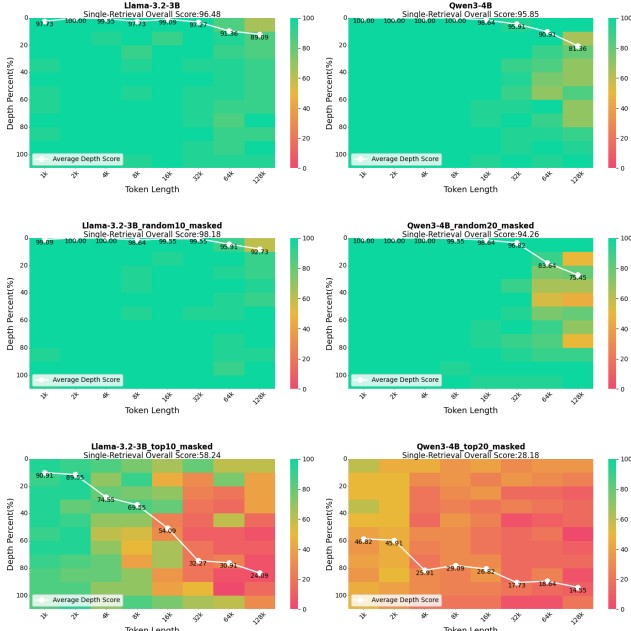

*Figure 3.* Needle-in-a-Haystack performance of Llama-3.2-3B-Instruct and Qwen3-4B under 128K context in the single-needle retrieval setting. Compared with randomly masking equal amount of attention heads, masking our Top-10 or Top-20 LPS attention heads leads to a clear performance drop on longer inputs.

randomly selecting attention heads. It is worth noting that some retrieval heads may not receive a higher potential score than other heads, but they still exhibit the expected kernel properties shown in Figure 2.

### 4.2. Behaviors of Long-context heads

After identifying long-context heads from the properties of static frequency kernels, we then perform actual inference on long-context inputs to further study the special behaviors exhibited by these attention heads.

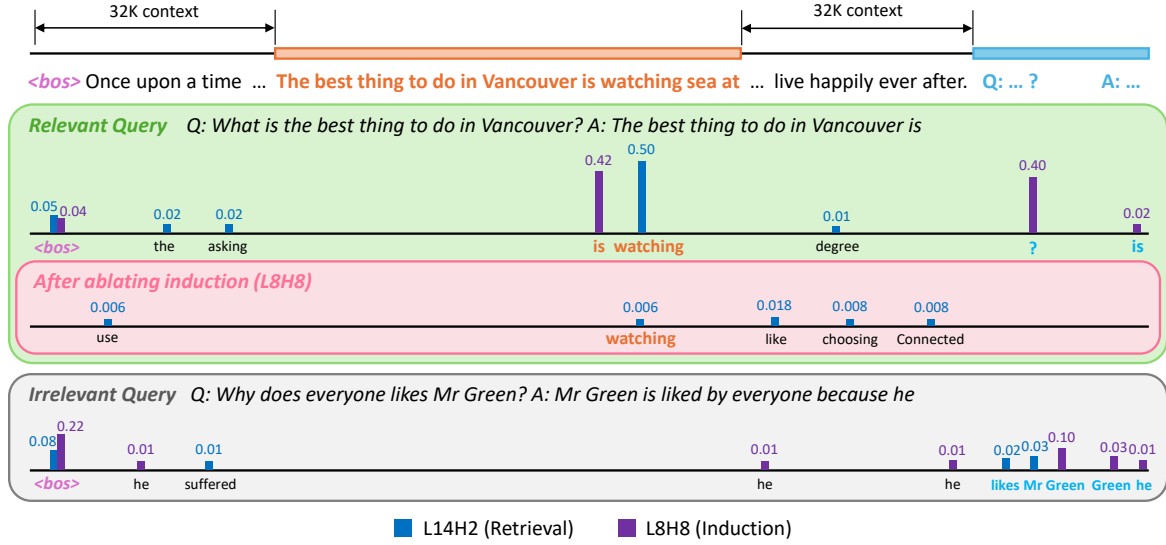

*Figure 4.* Softmax attention score distributions of different long-context heads for different queries in the NIAH scenario. The retrieval head strongly attends to the next token to be predicted. In contrast, the induction head always attend to tokens that are identical to the current input token. When removing the preceding induction head, the retrieval head sharply loses ability to attend to the expected token.

*Table 2.* The overlap of Top-$k$% identified attention heads using our retrieval potential score and "Retrieval score" (Wu et al., 2025) on different models.

| Top | Llama-3.2-3B-Instruct | | Qwen3-4B | |
|---|---|---|---|---|
| | # of Overlap | % of Overlap | # of Overlap | % of Overlap |
| 1% | 3 | 50.0% | 6 | 54.6% |
| 2% | 5 | 38.5% | 12 | 52.2% |
| 5% | 14 | 42.4% | 24 | 42.1% |
| 10% | 38 | 58.2% | 57 | 49.6% |

**Long-context behavior: Induction and Retrieval** We use the NIAH task to analyze potential heads. In this task, we insert a sentence that is unrelated to the topic into a long context and ask the model to recall its content at the end. By examining the attention scores of different attention heads over previous tokens, we mainly identified two types of long-context heads with clear behavioral patterns: *induction heads* and *retrieval heads*. Induction heads always attend to tokens in the text that are identical to the current input token, while retrieval heads tend to attend to tokens that are needed for later continuation. As shown in Figure 4, we apply LPS to all attention heads in Llama-3.2-3B-Instruct and take L8H8 (scores 2nd) and L14H2 (scores 5th) as the example. *Relevant query:* when we query needle-related information at the end of the text, both heads successfully give high attention to the tokens where the needle appears. However, unlike typical next-token prediction behavior, L8H8 does not attend to the needle token to be predicted ("watching"), but attends to the token that matches the current input token ("is") instead. *Irrelevant query:* When the context does not contain the needle to be retrieved, the retrieval head L14H2 spreads its attention across the context and assigns more

attention to the BOS token, remaining in a dormant state. In contrast, although L8H8 also sinks more attention to the BOS token, it continues to perform a induction behavior and broadly allocates attention to tokens in the document that are identical to the input token. *Ablating induction:* Although the induction head does not directly contribute to the next needle token prediction, it still plays an important role in locating key needle tokens and amplifying their features. This allows the attention in later layers to focus more on tokens in nearby positions. To verify this, we average the attention distribution of L8H8, so that it loses the ability to attend to identical tokens. By modifying the behavior of only this single attention head, the attention of L14H2 to the needle drops sharply. This result demonstrates the critical role of induction heads in the long-context processing.

Recalling the properties of high LPS heads shown in Figure 2: both retrieval heads and induction heads are first insensitive to long distances, which allows them to locate information positions in the preceding long context; then they use low effective-rank kernels $K$ to respectively perform highly specialized feature extraction on key information. The low-frequency kernel energy concentration and low effective rank properties are highly aligned with their respective functions.

**Low frequency contributes more, but not lowest frequency** While the above analysis studies the behavior of the whole attention heads, we can also analyze the attention behavior at each frequency using frequency kernel components $K_f$. For some attention heads identified by LPS, we analyze the contribution of frequency kernels to the needle across different heads and different distances, as shown in

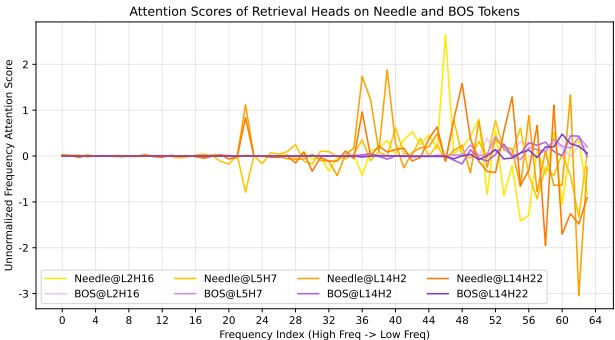

*(a)* Frequency contribution to needle and BOS tokens of 4 top retrieval heads. Relative distance to the needle is fixed at $n = -32k$.

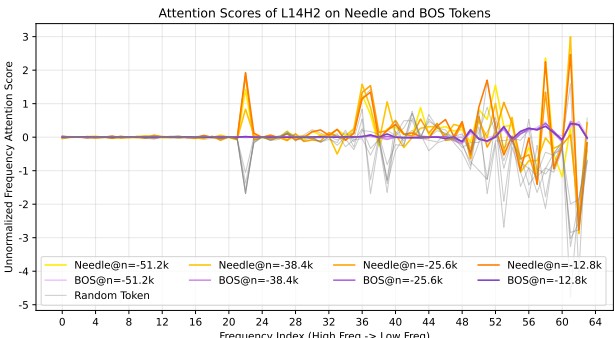

*(b)* Frequency contribution to needle and BOS tokens of L14H2 at different relative distance to the needle. Despite various $n$, the same frequency kernel often produces similar attention contributions. Gray lines indicate attention contributions to some random tokens in the context.

*Figure 5.* Frequency contribution to the final attention score on identified retrieval heads in the NIAH scenario shown in Figure 4.

Figure 5. We observe that these retrieval heads share some common contribution patterns: in the high-frequency band with low energy, the retrieval response is almost zero. In the low-frequency band where the energy is concentrated, different frequencies start to provide generally positive attention contributions to the needle, which makes the overall attention of the head to the needle token stand out. Consistent with the property that long-context heads concentrate their energy in low-frequency kernels, the positive contributions in these low-frequency bands can be stably amplified, thereby enabling a retrieval behavior.

Notably, in the last few low-frequency components (index 48-63 for example), contributions to the needle show strong oscillations. In this case, the response of $\boldsymbol{K}_f$ to the needle token becomes closer to its response to other random context tokens, as indicated by the gray line in Figure 5b. In contrast, for these extremely low-frequency components, the response of $\boldsymbol{K}_f$ to the BOS token begins to show stable and positive contributions. From this perspective, the lower-frequency kernels of long-context heads are mainly re-

sponsible for performing long-context functions. In contrast, their extremely low-frequency kernels act more like a stable "BOS extractor" and consistently allocate a certain amount of attention to the BOS (first) token. This phenomenon is consistent with the findings that RoPE and its low-frequency components are responsible for producing attention sinks (Gu et al., 2025; Xiong et al., 2025).

## 5. Amplifying Frequency Kernels Improves Long-context Performance

As attention heads with high long-context potential scores have favorable properties for processing long contexts, we assume that strengthening these properties can further improve model's long-context capabilities. Using the discoveries from previous frequency kernel analysis, in this section, we propose a simple static frequency parameter modification method to directly improve the model's long-context performance.

**Ablating induction heads from LPS** As identified in Figure 4, the top heads detected by long-context potential score may exhibit retrieval or induction properties. Although induction heads are also important in long-context scenarios, we find that modifications on these heads does not effectively improve long-context performance in practice. Therefore, we attempt to ablate induction heads from the computation of LPS first, so that we can focus more on heads with high retrieval potential.

Starting from the property of induction heads, they "always attend to tokens identical to the current input token". For the input hidden of query $\boldsymbol{x}_q$ and key $\boldsymbol{x}_k$, the induction kernel response $\boldsymbol{x}_q \boldsymbol{K} \boldsymbol{x}_k^\top$ should have a larger value when $\boldsymbol{x}_q$ and $\boldsymbol{x}_k$ are highly similar (inputting the same token). Based on this, we compute the symmetry score of a head's kernel:

$$s_{\mathrm{sym}} = \left| \frac{\|\boldsymbol{K}_{\mathrm{sym}}\|_F^2}{\|\boldsymbol{K}\|_F^2} - 0.5 \right|, \boldsymbol{K}_{\mathrm{sym}} = \frac{\boldsymbol{K} + \boldsymbol{K}^\top}{2}. \quad (11)$$

A kernel with a higher symmetry score has stronger self-correlation, which allows it to better match similar query and key hiddens. In practice, the symmetry score of retrieval head kernels is close to 0, while the symmetry score of induction head kernels can reach around 0.3.

By subtracting a certain proportion of $s_{\mathrm{sym}}$ from LPS, we can obtain the Retrieval Potential Score (RPS):

$$s_{\mathrm{RPS}} = s_{\mathrm{LPS}} - \beta s_{\mathrm{sym}}, \quad (12)$$

where $\beta$ is the coefficient to control the removal strength. Typically, setting $\beta = 0.5$ is enough to filter out induction heads from other long-context heads. It is notable that the computation of RPS is also fully static and can be done quickly without any model inference.

*Table 3.* Head amplification experiments on RULER and general datasets.

| Model | RULER | | | | | | General | | | | |
|---|---|---|---|---|---|---|---|---|---|---|---|
| | 4K | 8K | 16K | 32K | 64K | 128K | MMLU | HellaSwag | ARC-C | OBQA | GSM8K |
| **Llama-3.2-3B-Instruct** | 92.79 | 84.72 | 83.67 | 79.47 | 71.07 | **65.78** | **62.14** | 63.15 | 74.24 | 77.00 | **76.95** |
| - amplify 10 random heads | 92.97 | **85.24** | 83.53 | 78.69 | 69.93 | 64.38 | 61.99 | 62.76 | 74.24 | 77.60 | 75.59 |
| - amplify 10 top LPS heads | 93.54 | 84.78 | **84.03** | **79.66** | **71.58** | 64.66 | **62.14** | **63.20** | **75.25** | **78.00** | 74.75 |
| **Llama-3.1-8B-Instruct** | **96.18** | 94.23 | 92.28 | 85.50 | 84.48 | 75.81 | **69.06** | 76.83 | 83.05 | 80.60 | **84.15** |
| - amplify 20 random heads | 95.79 | 94.10 | 90.64 | 85.32 | 84.25 | **76.45** | 68.17 | 75.75 | **84.41** | 79.60 | 81.96 |
| - amplify 20 top LPS heads | 96.10 | **94.57** | **93.42** | **87.00** | **85.56** | 76.20 | 68.61 | **76.87** | 83.39 | **81.2** | 82.87 |
| **Qwen3-4B** | **95.08** | 91.47 | 89.64 | 86.85 | 75.66 | 66.30 | **66.69** | 77.83 | **82.71** | **78.00** | **42.15** |
| - amplify 20 random heads | 93.99 | 91.16 | 89.89 | 84.18 | 71.75 | 61.11 | 65.62 | 76.15 | **82.71** | 75.20 | 40.71 |
| - amplify 20 top LPS heads | **95.08** | **92.32** | **90.07** | **86.90** | **77.07** | **68.41** | 66.60 | **77.95** | 80.00 | 77.60 | 41.24 |

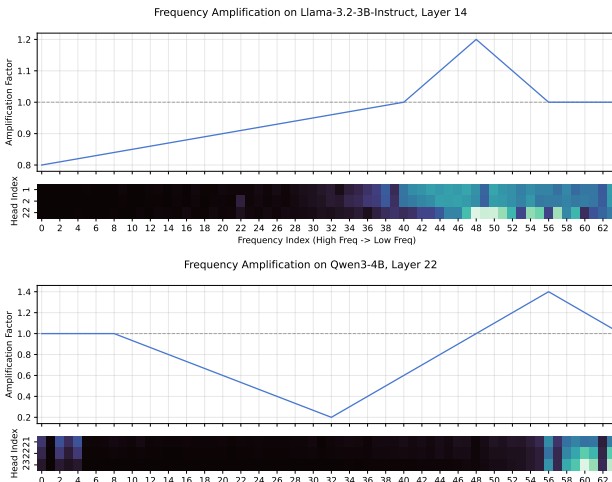

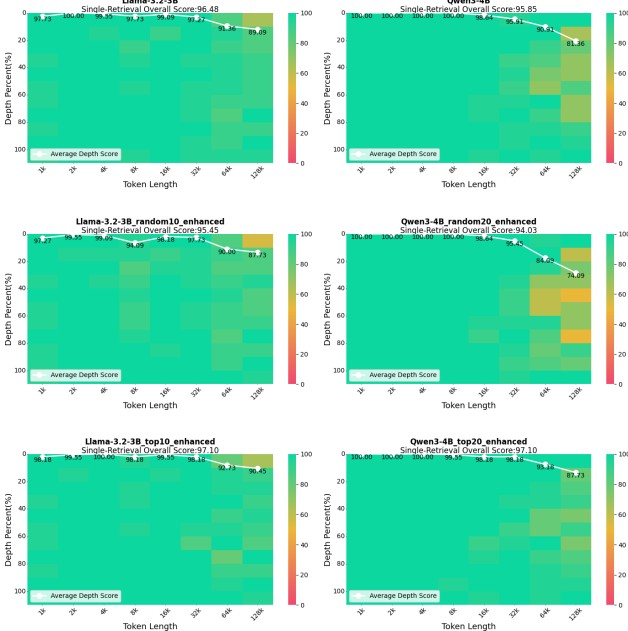

*Figure 6.* Amplification factors of different frequencies for Llama-3.2-3B-Instruct and Qwen3-4B. To improve model performance on long-context tasks, we design factor schedulers based on the frequency kernel energy distributions of different models. However, the core idea is always to amplify low-frequency components and suppress irrelevant high-frequency components.

**Enhancement on potential retrieval heads** Inspired by Figure 5, if we amplify the positive contribution of low-frequency kernels to the key token, the attention head will focus more on this token. To further enhance the long-context capability of potential retrieval heads, we adopt a strategy to suppress high frequencies and amplify low frequencies. Specifically, for a frequency index $f$ and its corresponding kernel $\boldsymbol{K}_f = \boldsymbol{q}_f^\top \boldsymbol{R}_f \boldsymbol{k}_f$ (Equation 5), we assign a scalar amplication factor $c_f$ to its $2 \times h$ frequency block $\boldsymbol{q}_f$, thus the modified kernel $c_f \boldsymbol{K}_f$ would have $c_f^2$ times of energy than before. By adjusting the value of $c_f$ with respect to the frequency index $f$, this strategy can further reduces the energy proportion of high-frequency components and increases the contribution of low-frequency bands, finally improving the LPS.

In practical static frequency kernel analysis, different models may have different energy distributions in the frequency

*Figure 7.* After amplification, the Needle-in-a-Haystack performance of Llama-3.2-3B-Instruct and Qwen3-4B under 128K context in the single-needle retrieval setting. Amplifying our top RPS heads could yield performance improvements on longer contexts.

spectrum. As shown in Figure 6, the factor scheduler we used in Llama-3.2-3B-Instruct aligns well with the low-frequency components we want to enhance in Figure 5, without amplifying the very low-frequency oscillation noise or the BOS token. However, in Qwen3-4B, the retrieval heads show non-negligible energy accumulation in the high-frequency band, while most of the energy is concentrated in much lower frequencies.

**Experiments** We use the same long-context datasets and model settings in the masking experiment of §4.1. Using the amplification factor scheduler in Figure 6, we amplify 10 top RPS heads in Llama-3.2-3B-Instruct (around $1.5\%$ of total heads), 20 top RPS heads in Llama-3.1-8B-Instruct (around $2\%$) and 20 top RPS heads in Qwen3-4B (around $1.7\%$).

Experiment results are shown in Figure 7 and Table 3. On the NIAH task, amplifying the low-frequency components of retrieval heads effectively improves the model's retrieval ability at extremely long 64K and 128K distances. For more complex long-context tasks in RULER, these enhancements can also bring better performances in most cases. As enhancing only such a small number of attention heads can lead to significant improvements, this result directly demonstrates the importance of low-frequency energy concentration for retrieval performance. Finally, since our frequency amplification method is entirely based on static analysis of parameter properties, this long-context enhancement method is very fast and easy to use.

## 6. Conclusion

In this paper, we propose the Long-context Potential Score (LPS) to measure the potential of attention heads to process long contexts. This metric is based on static analysis of frequency kernels formed by RoPE frequency components of attention heads. Experiments show that the attention heads identified by LPS in a static manner indeed have strong long-context modeling capability. In addition, by analyzing the head kernels and frequency kernels of these high-potential heads, we can further improve long-context performances of the model using a simple amplification strategy. This study of LLM parameter properties provides a static perspective on the functionality and interpretability of attention heads. We hope that more research will appear in the future, allowing functions in attention heads to be identified without actually running the model.

## Acknowledgments

This work was supported by the China NSFC Projects (62576212, 62120106006, 92370206, U23B2057).

## Impact Statement

This paper presents work whose goal is to advance the field of Machine Learning. There are many potential societal consequences of our work, none which we feel must be specifically highlighted here.

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
