# OpenReview forum: "Revealing Long-context Potential of Attention Heads via Frequency Kernels"
_ICML.cc/2026/Conference — ICML 2026 regular_

### Official Review · Reviewer_KEHp · 2026-03-05

**Soundness:** 2
**Presentation:** 3
**Significance:** 2
**Originality:** 3
**Overall Recommendation:** 4
**Confidence:** 3

**Summary:**

This paper addresses the challenge of identifying long-context-critical attention heads in LLMs, where existing methods rely on costly long-text inference. The authors address the concept of analyzing heads via static parameters: decomposing RoPE-based attention kernels into frequency components, they find effective long-context heads exhibit concentrated low-frequency energy and low effective rank. Building on this, they propose the inference-free LPS metric to quantify long-context potential, validate it on two 3B/4B models, and design a lightweight low-frequency amplification strategy for performance enhancement.The principal contribution consists of a static frequency kernel analysis framework, the LPS metric for efficient head identification, and a plug-and-play static enhancement strategy—opening a new direction for low-resource long-context optimization.

**Compliance With Llm Reviewing Policy:**

Affirmed.

**Final Justification:**

The author effectively addressed my most concerned issues of reproducibility and generalization by open source code, supplementing with 7B+and encoder decoder experiments, and acknowledging method limitations (such as ALiBi not being applicable). Although the theoretical causal explanation is not complete, empirical evidence is sufficient to support the core claim.

balance:

Originality and Importance: Strong. Static parameter analysis focuses on the potential of long texts without the need for inference, which is a new paradigm, and the low-frequency amplification strategy is lightweight and practical.

Clarity: Good. The paper writing is clear and the charts are intuitive.

Robustness: Moderate. The experimental coverage is limited (2 3B/4B models), but refutation supplements larger models and different architectures, and open-source code facilitates community validation.

Conclusion: Based on the positive feedback from the other three reviewers (Strong Accept, Accept, Weak Accept), I believe that this work has made significant contributions and is worthy of acceptance. My original Weak Reject has been adjusted to Accept.

**Key Questions For Authors:**

1. Theoretical Justification for Core Causal Claims

The paper only observes an empirical correlation between LPS (low-frequency energy concentration, low effective rank) and long-context capability, with no mathematical proof for the causal mechanisms. Could you supplement proofs for: (1) why low-frequency energy boosts long-context ability; (2) kernel effective rank-retrieval accuracy bounds; (3) amplification’s boundary conditions and numerical stability? Also analyze LPS’s basic mathematical properties (continuity, differentiability) and effective rank estimation variance.These derivations are critical to justifying the method’s theoretical rigor.

2. Generalizability and Hyperparameter Robustness

All experiments are limited to two 3B/4B RoPE models (no Q/K bias). Could you validate the method on alternative architectures (ALiBi, encoder-decoder), 7B+ models, and long-context fine-tuned models? Provide sensitivity analysis for hyperparameters α, β, and n via ablation studies.Impact on evaluation: This determines if the method is niche or broadly applicable, directly influencing assessments of its significance.

3. Reproducibility and Writing Compliance

The paper lacks open-source code and fails to declare inherited experimental settings (NeedleBench, Wu et al. (2025) Retrieval Score) or modifications. Could you release complete runnable code and explicitly declare these settings with citations?Impact on evaluation: Reproducibility is foundational—without code and transparent setup, the community cannot verify claims, undermining academic validity.

4. Rigor of Head Classification

Retrieval/induction head classification relies solely on two qualitative case studies. Could you provide quantitative stats (proportions, layer distribution) and ablation experiments to verify their distinct functional roles?Impact on evaluation: Quantitative evidence strengthens the classification’s rigor—without it, the interpretive insights remain speculative.

5. Amplification Side Effects and Failure Cases

The paper does not validate the amplification strategy’s impact on non-long-context tasks or explain the code completion performance drop (Table 3). Could you supplement side-effect experiments, clarify the performance drop, and discuss out-of-distribution failure cases?Impact on evaluation: Understanding side effects and failure boundaries is critical for practical utility—without this, real-world applicability remains uncertain.

**Limitations:**

The authors do not provide a dedicated discussion of the work’s core technical limitations. Key limitation is the lack of theoretical rigor: this paper only finds an empirical correlation between low-frequency energy concentration/low effective rank and long-context ability, with no mathematical proof for why these properties enable stronger long-context processing. These unaddressed technical constraints hinder a clear understanding of the method’s applicable scope and foundational robustness.

**Strengths And Weaknesses:**

Soundness:

Strengths: RoPE-based frequency kernel decomposition aligns with mainstream LLM architectures. Preliminary ablation and gold-standard overlap experiments confirm LPS’s empirical validity. The static amplification strategy is internally consistent.

Weaknesses: The core theoretical chain lacks mathematical rigor. The paper only observes an empirical link between low-frequency energy concentration/low effective rank and long-context ability, but provides no proof for why these properties necessarily enable stronger long-context processing. Key gaps include: no theoretical bound between kernel effective rank and retrieval accuracy, unproven mathematical properties of LPS (continuity, differentiability, convexity), and no analysis of the boundary conditions for low-frequency amplification to work (e.g., when it might fail) or proof of its numerical stability. Additionally, reproducibility is compromised by the absence of open-source code and incomplete implementation details, while experimental validation is narrow (limited to two small models), undermining the reliability of the claims.

Presentation:

Strengths: Well-structured with clear narrative flow; core contributions and technical details are coherently presented.

Weaknesses: Academic writing norms are not fully followed. The paper uses standard benchmarks (NeedleBench, LongBench) and the Retrieval Score method from prior work but fails to explicitly declare whether it adheres to the original experimental settings (prompt templates, evaluation pipelines, hyperparameters) or specifies any modifications. The comparison with Chiang & Yogatama (2025) (a closely related work on RoPE frequency analysis) is superficial, lacking clear delineation of research goals and methodological differences. Additionally, the initial definition of frequency index f (where larger f corresponds to lower RoPE frequency) is misaligned with the standard RoPE formula, which may confuse readers.

Significance:

Strengths: Offers a low-cost alternative to inference-based methods, with practical value for academic interpretability and industrial optimization. Opens a new static parameter analysis direction.

Weaknesses: This paper could be further discussed on models of more scales, such as 7B+ models or models that have undergone long-context fine-tuning/context window extension.

Originality:

Strengths: Novel static frequency kernel perspective for long-context head analysis; inference-free LPS metric is a meaningful methodological innovation.

Weaknesses: The core components build on established paradigms. Frequency analysis of RoPE and effective rank measurement are not new concepts in attention research; the primary novelty lies in integrating these elements into a static framework for long-context head identification, rather than introducing a fundamentally new theoretical breakthrough. The low-frequency amplification strategy, while practical, is a straightforward extension of the observed kernel properties, lacking innovative technical design beyond scaling frequency component magnitudes.

---

> ### Author Rebuttal · Authors · 2026-03-31
>
> Thank you for your thoughtful comments and suggestions on our work. As the text length is limited, we provide the results and figures here: https://anonymous.4open.science/r/long-context-heads-4E8A/README.md. The root of this anonymous repo is also our open-source code, so feel free to check our implementation details here!
>
> W1: To study the relationship between RPS and retrieval accuracy, we conduct a NIAH evaluation (see Fig. R1). The attention head retrieval accuracy and the RPS metric show a moderate positive correlation (r≈0.5). We also measure the next-token retrieval and current-token retrieval accuracy of the top retrieval and induction heads, and the results indicate that these top heads achieve high retrieval accuracy.
> For math properties of LPS: due to space constraints we provide a concise sketch of the proof. For a fixed relative distance n, the LPS is fully differentiable (and thus continuous) with respect to the parameter matrices $W_Q$ and $W_K$ by construction. The computation of LPS involves only linear transformations and elementary functions, all of which are differentiable in the defined domain. Consequently, the gradient can be propagated back to the weights via the chain rule. This differentiability implies that the LPS can theoretically serve as a differentiable regularization term to explicitly optimize attention heads for long-context capabilities during training.
> For the boundary conditions for amplification: we conducted a quick hyperparameter search on NIAH (see Fig. R4). The results show that different amplification factors lead to different performance outcomes, but they are generally better than or on par with the original model. We do observe cases where amplification fails in practice, which is mainly due to an inappropriate choice of the amplification range or the presence of bias terms in the attention (e.g., Qwen2.5).
>
> W2: We use OpenCompass (a library similar to lm-evaluation-harness) for evaluation. In the `opencompass` directory of our anonymous repo you can find the configuration files, which provide evaluation details such as the specific prompts and hyperparameters.
> Difference from Chiang et al: Their work also observes the impact of RoPE frequencies on long-context capability, making their motivation similar to ours. However, their method identifies heads with low dimension utility by training sparse masks (Sec. 5.2), whereas we rely on a purely static parameter analysis approach to detect long-context heads. Based on this, we further derive functionalities of these heads and propose methods to enhance long-context capability. Therefore our methodological and goals differ much from theirs.
>
> W3: We conduct more experiments on 7B+ models (including amplification impact on non-long-context tasks, and different variants), please see Fig. R5 & R6.
>
> W4: Thank you for recognizing the novelty of our static frequency kernel analysis! Indeed our method relies on existing tools, but the proposed LPS framework provides a practical way to shift long-context ability analysis from dynamic inference to static parameter analysis, representing a paradigm-level innovation. Regarding the low-frequency amplification strategy, we propose a direct application based on our empirical observations, but it may also has other potential uses (e.g. amplify low-frequency components during model initialization, allowing the model to acquire stronger long-context capabilities from the beginning of pretraining). We believe that our LPS framework is sufficiently general to influence future model design.
>
> Q1: We provided some answers in W1; here we just explain (1). As low-frequency kernels are less sensitive to distance, their results of QK^T are less affected by rotations. As a result, they can produce nearly constant responses across different distances, which is crucial for long-context capability. Enhancing the energy of low-frequency components effectively amplifies the influence of these kernels, increasing their contribution and thereby improving long-context performance.
>
> Q2: Broader model experiments are shown in W3. For alternative architectures: ALiBI is not well suited for this frequency kernel analysis, because its relative distance bias in attention does not act on the rows/columns of the $W_Q,W_K$, but rather on the final attention score. For encoder-decoder models (t5gemma-2b-2b), if RoPE is used, our method is still partially applicable (mainly in the self-attention layers, as cross-attention does not involve positional encoding). Removing top LPS heads in the encoder causes more performance drop on NIAH than decoder (see Fig. R7): it is because long-context tasks are primarily handled by the encoder when processing the input sequence, whereas the decoder is typically responsible for generating shorter outputs.
>
> Constrained by length we cannot resolve all your concerns here. Please let us know if you have other questions! Your suggestions make our work better.

---

> > ### Author Rebuttal · Reviewer_KEHp · 2026-04-02
> >
> > The author has open sourced the code, supplemented experiments such as 7B+, acknowledged the limitations of the method, and provided explanations. Based on the positive feedback from other reviewers, I have revised it to Accept.

---

> > > ### Author Response · Authors · 2026-04-02
> > >
> > > Thank you so much for recognizing our work! We sincerely appreciate your constructive review comments and the effort you devoted to the review.

---

### Official Review · Reviewer_YPX7 · 2026-03-11

**Soundness:** 4
**Presentation:** 4
**Significance:** 3
**Originality:** 2
**Overall Recommendation:** 5
**Confidence:** 4

**Summary:**

The paper proposes a static method to identify long-context attention heads in RoPE-based transformers. The authors decompose each head into frequency kernels induced by RoPE and define a Long-context Potential Score that favors low-frequency energy concentration and low effective rank. They show that masking high-score heads degrades long-context benchmarks and present a simple low-frequency amplification method that yields modest gains on retrieval-oriented long-context tasks.

**Compliance With Llm Reviewing Policy:**

Affirmed.

**Final Justification:**

The authors resolved my questions and concerns. I will keep my original score since it's already very positive.

**Key Questions For Authors:**

See Weaknesses.
I would appreciate experiments on larger models (at least 8B).

**Limitations:**

Yes

**Strengths And Weaknesses:**

S1 The paper studies attention heads from a frequency-domain perspective and provides a clear formulation that connects RoPE frequencies with long-context retrieval behavior.

S2 The proposed metric is static and does not require long-context inference, which makes the analysis computationally inexpensive.

S3 The masking experiments show that removing / enhancing top-scoring heads significantly harms / enhances long-context tasks, which supports the claim that these heads are functionally important.

S4 The presentation is nice and clean. The paper is generally easy to follow, and the figures help communicate the main intuition.

W1 Novelty is limited. The main contribution is a static scoring heuristic built on RoPE frequency kernels, but the broader ingredients are already well established in prior work. Retrieval heads, induction heads, kernel views of attention, and RoPE frequency analysis have all been studied before. Therefore, the contribution appears a bit incremental rather than substantial.

W2 The qualitative analysis in Figure 4 is suggestive but potentially cherry-picked. The authors visualize only two hand-selected high-LPS heads (L8H8 (scores 2nd) and L14H2 (scores 5th)), without a systematic analysis of how many top-ranked heads actually exhibit retrieval-like or induction-like behavior. As a result, the figure serves as an illustrative case study rather than strong evidence that these behavioral categories broadly characterize high-LPS heads.

W3 The scoring design remains somewhat heuristic. LPS combines low-frequency energy concentration and an entropy-based effective-rank term with manually chosen coefficients, but the manuscript provides limited justification for these specific design choices and does not clearly establish how sensitive the conclusions are to these settings.

W4 The model selection is somewhat tricky and weakens the empirical case. The paper uses Llama-3.2-3B-Instruct on one side and non-instruct version of Qwen3-4B on the other. This makes the setup look somewhat selective (maybe the version of instruct/non-instruct is cherry picked). In addition, both evaluated models are relatively small, so it is unclear whether the conclusions transfer to stronger and more practically relevant larger models.

W5 The empirical scope is still limited. The paper mainly demonstrates the method on two small models and a narrow set of analyses. Stronger evidence would require broader validation across more architectures, scales, and training variants.

---

> ### Author Rebuttal · Authors · 2026-03-31
>
> Thank you for your thoughtful comments and suggestions on our work. We provide the additional results and figures here: https://anonymous.4open.science/r/long-context-heads-4E8A/README.md. The root of this anonymous repo is also our open-source code, so feel free to check our implementation details here!
>
> W1. Very thanks for your recognition of our work. This study indeed builds upon prior research such as retrieval heads and RoPE frequency analysis (we sincerely appreciate these excellent works!). Nevertheless, we utilize these ideas to propose a new static method for analyzing the long-context capability of attention heads, and we further distinguish the functions of attention heads based on their parameter properties - few previous works have achieved such investigation. We believe that the static frequency kernel analysis method proposed in our paper can inspire more researchers to study and optimize the intrinsic properties of model parameters.
>
> W2. Figure 4 intends to provide an illustrative qualitative analysis to show the functions of different attention heads. Here, we further conduct quantitative experiments to analyze the behaviors of retrieval heads and induction heads, please see Fig. [R1](https://anonymous.4open.science/api/repo/long-context-heads-4E8A/file/rebuttal/R1_RPS_retrieval_relation.pdf). The attention head retrieval accuracy and the RPS metric show a moderate positive correlation (Pearson r≈0.5). We also measure the next-token retrieval and current-token retrieval accuracy of the top retrieval and induction heads, and the results indicate that these top heads achieve high retrieval accuracy. This would be a stronger evidence to characterize high-LPS heads.
>
> W3. We provide a search on hyperparameters $\alpha, \beta$ of the RPS, please see Fig. [R2](https://anonymous.4open.science/api/repo/long-context-heads-4E8A/file/rebuttal/R2_RPS_hyper_search.pdf). We perform a grid search to obtain top RPS heads under different settings, and then compare them with the golden "retrieval heads" (Wu et al., 2025) identified from actual inference. The results show that, within a reasonable $\alpha, \beta$ range, different hyperparameter combinations typically lead to less than a 10% difference in overlap. Apart from $\alpha, \beta$, we do more hyperparameter search (distance and low-frequency amplification), you may check Fig. [R2](https://anonymous.4open.science/api/repo/long-context-heads-4E8A/file/rebuttal/R2_RPS_hyper_search.pdf)-[R4](https://anonymous.4open.science/api/repo/long-context-heads-4E8A/file/rebuttal/R4_amplification_hyper_search.pdf) out.
>
> W4 & W5. Firstly, we want to clarify that both Llama-3.2-3B-Instruct and Qwen3-4B models used in our paper are instruct models (the non-instruct version of Qwen3-4B is named [Qwen3-4B-Base](https://huggingface.co/Qwen/Qwen3-4B-Base)). For experiments on larger models, we provide a full experiment result on Llama-3.1-8B-Instruct (see Fig. [R5](https://anonymous.4open.science/api/repo/long-context-heads-4E8A/file/rebuttal/R5_general_tasks_and_llama3.1_8b_results.pdf)), and some top-LPS head masking results on Qwen2.5-7B base/instruct/1M-context series (see Fig. [R6](https://anonymous.4open.science/api/repo/long-context-heads-4E8A/file/rebuttal/R6_qwen2.5_7b_series.pdf); since Qwen2.5 introduces bias terms in attention parameters, the formulas proposed in our paper are not directly applicable, so we only conduct head removal experiments). These additional experimental results are also well aligned with the conclusions of the paper, and we hope they can strengthen the generalizability of our conclusions.
> P.S. Regarding the empirical scope, we are currently exploring amplifying the low-frequency components of attention heads at the very begining of the pretraining stage (during initialization). Preliminary pretraining results on a 1B-scale model show that this approach can significantly improve long-context capabilities, while incurring only ~0.03 increase in PPL. We believe the findings of this paper can also help guide the training of better long-context models!
>
> Thank you again for your appreciation of our work! Please let us know if you have other questions.

---

> > ### Author Rebuttal · Reviewer_YPX7 · 2026-03-31
> >
> > Thanks for the rebuttal response. I've read through all of it.
> > I will keep my original score since it's already very positive.

---

> > > ### Author Response · Authors · 2026-04-02
> > >
> > > We are glad to have adequately addressed your concerns! Thank you again for your appreciation of our work and for your thoughtful review.

---

### Official Review · Reviewer_c7Kg · 2026-03-12

**Soundness:** 3
**Presentation:** 3
**Significance:** 3
**Originality:** 2
**Overall Recommendation:** 5
**Confidence:** 4

**Summary:**

The paper proposes a static, training- and inference-free method to identify attention heads with strong long‑context capability by analyzing “frequency kernels” induced by RoPE within each head. It proposed the Long-context Potential Score (LPS), which sums the frequency-energy distribution scores and the effective rank scores, to favor distance-insensitive low-frequency attention heads. The authors then validate the importance of the high-LPS heads for long-context tasks through head masking, showing nontrivial overlap with retrieval heads identified online, analyze induction vs. retrieval behaviors, and connect extremely low‑frequency bands to BOS attention sinks. Furthermore, the author improved the models’ performance on long-context tasks by selectively amplifying low-frequency attention heads with high LPS scores.

**Compliance With Llm Reviewing Policy:**

Affirmed.

**Final Justification:**

The rebuttal effectively addresses my concerns with additional experiments and improved clarity, strengthening both the empirical support and overall impact, so I increase my score.

**Key Questions For Authors:**

1. How robust are the LPS/RPS rankings to the choice of distance $n$ and hyperparameters $\alpha$ and $\beta$?

2. Can you provide a more principled derivation or empirical validation for the symmetry score as an induction proxy (e.g., correlation with online induction-head labels or with attention-to-identical-token hit rates)?

3. How does amplification affect perplexity and general benchmarks (MMLU, ARC, HellaSwag, GSM8K, etc.)? Are there measurable regressions or trade-offs?

**Limitations:**

yes

**Strengths And Weaknesses:**

## 1. Soundness

### Strengths
- The core idea of analyzing the model from a frequency distribution perspective is quite insightful. Using a decomposition method to rank different components is logically consistent.
- The use of LPS (Low-frequency Power Score) and RPS (Retrieval Potential Score) effectively bridges the gap between signal processing (spectral density) and transformer mechanics.
- Initial experiments—such as removing high-scoring components—show a clear impact on performance, which lends some empirical support to the paper’s claims.

### Weaknesses
- Some core metrics, such as the "effective rank," may exhibit slight discrepancies relative to standard mathematical definitions, which could lead to ambiguity.
- The theoretical derivation relies on several simplifying assumptions, such as overlooking potential interference between different frequency bands; this might lead to inaccuracies in specific data distributions.

## 2. Presentation

### Strengths
- The paper is well-structured, moving smoothly from theoretical analysis to tool development and final validation.
- The visualizations are clear and help the reader intuitively understand the functional differences among the model’s components.

### Weaknesses
- Certain critical details, such as the specific schedules for parameter adjustment, are described somewhat vaguely. This might make it difficult for other researchers to replicate the results.
- Some terms, such as "high" vs. "low" frequency, are used descriptively rather than operationally, lacking clear numerical thresholds.

## 3. Significance

### Strengths
- This method offers a way to identify key model components without requiring massive computational resources, which is valuable for research environments with hardware constraints.
- The exploration of how frequency concentration relates to long-context capabilities offers a helpful perspective on model behavior.

### Weaknesses
- The current research is primarily limited to a specific type of encoding; its generalizability to other mainstream architectures remains to be seen.
- While it improves performance in specific areas, the paper does not provide sufficient evidence that these interventions cause regressions in basic tasks or general reasoning capabilities.

## 4. Originality

### Strengths
- Combining spectral analysis with a static ranking metric for component selection is a creative approach.
- The introduction of a "symmetry score" to distinguish between different functional behaviors provides an interesting new angle for mechanistic analysis.

### Weaknesses
- While the perspective is fresh, the comparison with existing similar works is not deep enough.
- Based on the current results, it is difficult to determine if this proposed method is significantly superior to existing "training-free" fixes that also manipulate frequency components.

---

> ### Author Rebuttal · Authors · 2026-03-31
>
> Thank you for your thoughtful comments and suggestions on our work. As the text length is limited, we provide the results and figures here: https://anonymous.4open.science/r/long-context-heads-4E8A/README.md. The root of this anonymous repo is also our open-source code, so feel free to check our implementation details here!
>
> W1: Our "effective rank" is defined as Eq. (6) in the paper. Though it's different from the definition in Roy & Vetterli (2007), our method still adopts the core idea of the effective rank (using Shannon entropy to measure the unevenness of a distribution) to quantify the energy imbalance across different frequency components. Another advantage of our definition is that computing matrix energy is straightforward, and it avoids performing heavy SVD on large parameter matrices as in the original definition. This makes our static parameter analysis more efficient. As to the interference among frequency bands, because different frequency components of RoPE tend to be asymptotically orthogonal in high-dimensional embedding spaces (like h=3072 in Llama-3.2-3B), it's usually safe to ignore the slight overlap between frequencies.
>
> W2. We provide the anonymous code repo here, and will make it public to others for reproducibility! For terms like "high" or "low", we indeed intended to provide a relative description rather than an absolute categorization. In our empirical observations, "high" frequency components tend to fall within top ~25% of the frequency components, whereas the "low" frequency components that are more responsible for long-context tend to concentrate in the bottom ~35%.
>
> W3. Our method is indeed more suitable for LLMs based on RoPE. Nevertheless, given that RoPE-based models are still leading the current mainstream, we believe that our frequency kernel analysis approach is already impactful.
> Regarding the effect of low-frequency amplification on general tasks: we have added additional experiments, please see Fig. R5. Results show that applying low-frequency amplification to the top LPS heads may either improve or degrade performance on general tasks, but the overall impact is not significant. Moreover, it often performs better than amplifying the same number of random attention heads. This suggests that low-frequency enhancement is sufficiently robust for general tasks as well.
>
> W4. Currently, there is limited work that analyzes the long-context capability of attention heads using static frequency kernels. The most closely related work to ours is Chiang & Yogatama (2025): their work also observes the impact of RoPE frequencies on long-context capability, making their motivation similar to ours. However, their method identifies heads with low dimension utility by training sparse masks (Sec. 5.2), whereas we rely on a purely static parameter analysis approach to detect long-context heads. Based on this, we further derive functionalities of these heads and propose methods to enhance long-context capability.
> Other approaches that manipulate frequency components include NTK-aware methods and YaRN, which extend the maximum inference length by modifying rotational frequencies. In contrast, our low-frequency amplification method improves long-context performance by adjusting the amplitude of different frequency components in the parameters. These approaches are orthogonal and can be combined! In fact, in our experiments, Qwen3 uses YaRN for 128k-length inference, and we further apply our enhancement on top of YaRN, achieving improved long-context performance.
>
> Q1. We provide a search on distance $n$ and hyperparameters $\alpha, \beta$ in Fig. R2&R3. Overall, the choice of the relative distance $n$ has little effect on the LPS scores for most attention heads; only the attention heads in layer 0 are sensitive to this distance. For $\alpha, \beta$ of the RPS score, we perform a grid search to obtain top RPS heads under different settings, and then compare them with the golden "retrieval heads" (Wu et al., 2025) identified from actual inference. The results show that, within a reasonable $\alpha, \beta$ range, different hyperparameter combinations typically lead to less than a 10% difference in overlap.
>
> Q2. We conduct a NIAH evaluation across 40 64k long contexts on top induction & retrieval heads (Fig. R1). For top induction heads, they evidently attend to the current (identical) token more than other heads, validating the symmetry score as an induction proxy:
>
> | Success rate | All heads avg. | Top-5 retrieval heads avg. | Top-5 induction heads avg. |
> |-|-|-|-|
> |Next token|0.117|**0.990**|0.025|
> |Current token|0.103|0.155|**0.620**|
>
> \* "Success rate" is defined as: the target token is among the top-10 attention scores of the head
>
> Q3. For the performances of amplified models on general tasks, please check our response in W3 (Fig. R5)!
>
> Thank you again for your work in reviewing our paper, and we are here to address your questions if you have more concerns!

---

### Official Review · Reviewer_siPJ · 2026-03-19

**Soundness:** 4
**Presentation:** 3
**Significance:** 3
**Originality:** 4
**Overall Recommendation:** 6
**Confidence:** 3

**Summary:**

This paper identifies long-context heads by analyzing static frequency kernels and effective rank. It proposes the LPS metric and a training-free amplification strategy that significantly enhances retrieval performance without exhaustive inference, offering a highly efficient and interpretable approach to enhancing models' long-context capabilities.

**Compliance With Llm Reviewing Policy:**

Affirmed.

**Final Justification:**

The rebuttal addressed all my concerns. So I raised my score to 6.
But Reviewer KEHp points out some possible math problems of this paper which I do not have absolute confidence, so I will downgrade my confidence to 3.

**Key Questions For Authors:**

Is your work effective only for RoPE-based models? This is merely out of my own curiosity and does not affect the significance of your work, as RoPE has already become the de facto absolute mainstream.

**Strengths And Weaknesses:**

**Strength**

1. This paper investigates a meaningful topic by providing a deep analysis of the intrinsic nature of previously discussed long-context retrieval heads. Furthermore, based on this analysis, they propose a feasible and effective scheme for model enhancement. Techically solid.

2. The mathematical foundation of this paper is solid. The analysis of rotation amounts for different frequencies in RoPE is relatively mature, and the authors naturally connect this with retrieval heads. Furthermore, by incorporating effective rank to quantify the selectivity of attention heads, the theoretical logic remains self-consistent.

3. The enhancement method is a plug-and-play strategy. It improves long-context performance without the need for expensive fine-tuning, demonstrating high practical utility.

**Weakness**

1. While the authors demonstrate through experiments that removing retrieval heads has a negligible impact on the model's performance in normal tasks, they do not provide direct evidence showing that applying the enhancement method also maintains performance in those same scenarios. Relying solely on verbal inference for this conclusion lacks empirical rigor.

2. The author only conducts experiments on 4B-size models. Since the proposed enhancement method is a plug-and-play method, I believe that requiring the authors to provide experimental results on a 7B-scale model is not an unreasonable request.

3. The paper lacks an ablation study on the amplification factors. It is unclear how sensitive the model's performance is to different scaling intensities, or whether there exists an optimal range for these coefficients across different architectures.

---

> ### Author Rebuttal · Authors · 2026-03-31
>
> Thank you for your thoughtful comments and suggestions on our work. We provide the additional results and figures here: https://anonymous.4open.science/r/long-context-heads-4E8A/README.md. The root of this anonymous repo is also our open-source code, so feel free to check our implementation details here!
>
> W1. We additionally report the performance of the model on general tasks after applying the enhancement methods, please see Fig. [R5](https://anonymous.4open.science/api/repo/long-context-heads-4E8A/file/rebuttal/R5_general_tasks_and_llama3.1_8b_results.pdf). Results show that applying low-frequency amplification to the top LPS heads may either improve or degrade performance on general tasks, but the overall impact is not significant. Moreover, it often performs better than amplifying the same number of random attention heads. This suggests that low-frequency enhancement is sufficiently robust for general tasks as well.
>
> W2. You said right, so we add full experiments on Llama-3.1-8B-Instruct and some top-LPS head masking experiments on Qwen2.5-7B base/instruct/1M-context series (Fig. [R5](https://anonymous.4open.science/api/repo/long-context-heads-4E8A/file/rebuttal/R5_general_tasks_and_llama3.1_8b_results.pdf)-[R6](https://anonymous.4open.science/api/repo/long-context-heads-4E8A/file/rebuttal/R6_qwen2.5_7b_series.pdf)). These additional experimental results are also well aligned with the conclusions of the paper, and we hope they can strengthen the generalizability of our conclusions.
>
> W3. For the amplification factors, we conduct a quick hyperparameter search on Llama-3.2-3B-Instruct on NIAH task (see Fig. [R4](https://anonymous.4open.science/api/repo/long-context-heads-4E8A/file/rebuttal/R4_amplification_hyper_search.pdf)). The results show that the sensitivity does exist (different amplification factors lead to different performance outcomes), but they are generally better than or on par with the original model.
>
> Q1. From the idea presented in the paper, our method is best applicable to models based on RoPE. For models with non-RoPE positional encoding (taking absolute positional encoding as an example), the positional embeddings are directly added to the input hidden states. This additive form introduces additional complexity ( $Q_iK_j^\top=\left((x_i+p_i)W_Q\right)\left((x_j+p_j)W_K\right)^\top=x_iW_QW_K^\top x_j^\top+x_iW_QW_K^\top p_j^\top+p_iW_QW_K^\top x_j^\top+p_iW_QW_K^\top p_j^\top$ , there are four independent interaction terms, leading to highly complicated interactions), which cannot be directly modeled by Eq. (4) in our paper. A similar issue arises in attention computations with bias terms (e.g., the Qwen2.5 series, where the QK computation includes a bias): although we can still perform ablation experiments by removing high-LPS heads, the effectiveness of enhancement methods may degrade due to the interference introduced by the additional bias terms.
> For other positional encoding schemes, the situation is also different. For attention mechanisms such as ALiBI (mentioned by reviewer KEHp), the positional encoding term acts as a bias directly on the entire attention head, so the notion of a "frequency kernel" does not apply. A similar limitation exists for NoPE-based attention, where the rotation matrices $R(n)$ for all frequencies reduce to identity matrices, making frequency kernel analysis inapplicable.
> At present, our conclusions are applicable to most RoPE-based models, but we will also attempt to extend these findings to models with other architectures in future work.
>
> Thank you again for your appreciation of our work! We are here to address your questions if you have more concerns.

---

> > ### Author Rebuttal · Reviewer_siPJ · 2026-04-02
> >
> > The authors have satisfactorily addressed all of my concerns.
> >
> > At the same time, I notice that Reviewer KEHp has provided a more in-depth critique of the paper's content on RoPE frequency analysis. While I find the authors' exposition of the theoretical components intuitively convincing, my own familiarity with this area is limited, so I am not well-positioned to rigorously evaluate this aspect.
> >
> > All in all, based on my current knowledge, I maintain my previous assessment that this paper is highly solid both in terms of theoretical intuition and methodological effectiveness. Since the authors have resolved the concerns raised in my review, I am happy to raise my score to 6.

---

> > > ### Author Response · Authors · 2026-04-02
> > >
> > > We are truly delighted that we were able to fully address your concerns! Thank you again for your appreciation of our work and for your effort in review.

---

### Decision · Program_Chairs · 2026-04-30

**Decision:**

Accept (regular)

**Comment:**

All reviewers are positive about the work. Therefore, I am glad to recommend an acceptance.